# Do General Practitioners in a Visiting Medical Officer Arrangement Improve the Perceived Quality of Care of Rural and Remote Patients? A Qualitative Study in Australia

**DOI:** 10.3390/healthcare10061045

**Published:** 2022-06-04

**Authors:** I Nyoman Sutarsa, Rosny Kasim, Ben Steward, Suzanne Bain-Donohue, Claudia Slimings, Sally Hall Dykgraaf, Amanda Barnard

**Affiliations:** 1Rural Clinical School, Medical School, College of Health and Medicine, The Australian National University, Acton, ACT 2601, Australia; rosny.kasim@anu.edu.au (R.K.); ben.steward@anu.edu.au (B.S.); suzanne.bain-donohue@anu.edu.au (S.B.-D.); claudia.slimings@anu.edu.au (C.S.); sally.hall@anu.edu.au (S.H.D.); amanda.barnard@anu.edu.au (A.B.); 2Department of Population Health and Preventive Medicine, Faculty of Medicine, Udayana University, Denpasar 80232, Bali, Indonesia

**Keywords:** rural health, primary care, health service delivery, quality of care, qualitative study, Australia

## Abstract

Background: In rural and remote Australia, general practitioners (GPs) provide care across the continuum from primary to secondary care, often in Visiting Medical Officer (VMO) arrangements with a local hospital. However, little is known about the role of GP-VMOs in improving the perceived quality of care and health outcomes for rural and remote communities. Methods: We collected qualitative data from three GP-VMOs (all aged >55 years) and 10 patients (all aged over 65 years) in three local health districts of New South Wales, Australia. Thirteen in-depth interviews were conducted between October 2020 and February 2021. We employed thematic analysis to identify key roles of GP-VMOs in improving the perceived quality of care and health outcomes of rural and remote patients. Results: Our study advances the current understanding regarding the role of GP-VMOs in improving the perceived quality of services and health outcomes of rural and remote patients. Key roles of GP-VMOs in improving the perceived quality of care include promoting the continuity of care and integrated health services, cultivating trust from local communities, and enhancing the satisfaction of patients. Conclusions: GP-VMOs work across primary and secondary care creating better linkages and promoting the continuity of care for rural and remote communities. Employing GP-VMOs in rural hospitals enables the knowledge and sensitivity gained from their ongoing interactions with patients in primary care to be effectively utilised in the delivery of hospital care.

## 1. Introduction

Australians living in rural and remote areas have a shorter life expectancy and poorer access to and use of health services compared with people living in metropolitan areas [1]. Limited access to quality and timely health services is one of the fundamental causes of inequitable health outcomes between people living in major cities and rural and remote areas [2]. In New South Wales, cardiovascular disease hospitalization rates in 2019/2020 were lower in major cities than they were in remote areas (1561.9 and 2578.8 per 100,000 population). Similarly, death rates from all causes (per 100,000 population) were higher in remote areas (681.3) and very remote areas (625.2) than in major cities (449.5). Emergency department presentations per 100,000 population were higher in outer regional and remote areas (64,409.7) compared to major cities (29,819.2) [3]. 

Central to this accessibility issue is shortages of health staff and inequitable distribution of health resource between urban, rural, and remote areas of Australia [4,5]. The rates of employed full-time equivalent (FTE) clinicians in Australia decreases with increasing remoteness; and the gap has widened since 2013 [6]. A major contributor to these shortages is the difficulty in recruiting and retaining a skilled health workforce in sufficient number to sustain quality of health services in rural and remote areas [2,7,8]. These staff shortages have led to longer waiting times for diagnosis and treatment [9], reduced patient confidence in health providers [10], or the closure of hospital emergency departments [10].

Various studies have identified issues associated with the retention of health professionals in rural and remote locations. A qualitative review of six studies in Australia and Canada found that doctor retention in remote areas are influenced by peer and professional support, working culture and burnt out, personal issues, including family and time-off, organizational support, and cultural and gender issues [11]. 

Furthermore, the decisions of health workers to practice rurally are informed by non-financial incentives, including job satisfaction, career advancement opportunities, and professional development [12]. Other studies have identified strategies to sustain rural health workforce, including recruiting more doctors with rural background [13,14,15], longer rural immersions and regional training [14,15,16], and enhancing rural training opportunities and sustainable rural working condition for overseas trained doctors [17].

Hospitals in rural and remote areas are commonly reliant upon primary care doctors or other contracted health providers to meet the health needs of rural and remote communities, including for emergency care [18]. In the United States, about 67% of emergency clinicians at rural hospitals are non-emergency physicians, and are typically family physicians or GPs [18]. In Australia, it is hoped that many of these doctors will be rural generalists who have advanced skills in emergency, anaesthetics, or obstetrics in addition to their GP training [19]. However, sustaining this critical rural workforce, and valuing its unique contribution to service delivery may be problematic [20,21].

In rural and remote Australia, GPs may provide care across the continuum from primary to secondary care, often in visiting medical officer (VMO) arrangements with a local hospital. VMOs are medical practitioners appointed by the hospital to provide medical care to admitted patients on a sessional or fee-for-service basis [22]. GP-VMOs provide a range of medical and surgical services necessary to meet rural healthcare needs, especially for acute and emergency care, where access to other medical specialists is limited or non-existent [23]. Employing GPs in hospital care can mitigate overcrowding of emergency departments (ED) and reduce hospitalisation rates [24].

Improved access across the continuum of care, including access to hospital care through GP-VMOs, can contribute to improve health outcomes of rural and remote communities [25]. Increased continuity of care provided by GP-VMOs has been associated with lower mortality rates [26]. Relationships characterised by trust, empathy, and loyalty that exist between patients and their GPs can facilitate continuity of care [27], and may consequently improve the health outcomes of rural and remote communities. In addition to these interpersonal dimensions, GP-VMOs bring their clinical knowledge of patients, social understanding, and cultural competence acquired in the primary care setting, into the hospital environment to inform clinical decision making. 

GPs working in rural hospitals add value in improving clinical leadership and clinical competencies [28]. They may act as health provider, advocate, and communicator, especially for patients with complex health conditions [25,29]. Employing GPs in hospital care, through VMO arrangements, may provide more comprehensive and resource-effective care for ED patients presenting with non-urgent health problems [24], and can reduce patient waiting times and lengths of stay in the ED and hospital, subsequently freeing bed-days for patients with more critical health conditions [24]. 

Additionally, task-shifting multiple health services from a junior doctor to trained nurses in collaboration with existing local GPs has been found to be a cost-effective strategy in rural areas where the availability of GPs is limited [30]. Little is known about the specific role of GP-VMOs in improving the quality of care, patient safety, health outcomes, and satisfaction of rural and remote communities in Australia. Our recent scoping review [31] identified only four studies (out of 12) that had examined the effects of employing GPs in Australian rural and remote hospital care [10,25,32,33]. Two of these studies used medical record review and hospital data analysis [32,33], one was a case study [10], and one was a qualitative study exploring GP roles in colorectal cancer management [25]. 

Other studies were conducted in the UK, Ireland, the US, New Zealand, Canada and Nepal [31]. Our scoping review identified both positive and negative effects. The benefits included: (a) improved coverage of acute care management, (b) enhanced continuity of care, (c) improved utilisation of a health algorithm, (d) reduced waiting times, (e) decreased complication rates and referral to tertiary hospitals, (f) greater satisfaction of patients and providers, (g) prevention of ED from closures, and (h) improved skills and competence among health providers. The reported negative consequences included increases in prescribing and poor documentation of clinical history and physical examination [31].

Despite these studies, a substantial gap in the literature is the lack of rural and remote GP-VMO and patient perspectives regarding perceived GP-VMO roles and their relationship to perceived quality and health outcomes. GPs gain rich insights about patient circumstances from their ongoing interaction in primary care settings, and they bring these insights into hospital care within an integrated community-primary–secondary care model. 

This model primarily focuses on providing coordinated care for patients and their families, seamless integration across primary and secondary health services, and better linkages between health services, age care systems, and other social care systems [34]. How these insights, knowledge, and relationships translate into better quality of care for rural and remote patients remains under-investigated. This qualitative study examined how clinical, social, and cultural knowledge gained by GPs through primary care delivery influences the provision and perceived quality of secondary care services for rural and remote communities.

## 2. Methods

Study setting, design, and data collection: A qualitative study was conducted between October 2020 and March 2021, involving selected rural and remote locations in three local health districts (LHDs) of rural New South Wales (NSW), Australia: Hunter New England, Southern NSW, and Western NSW. These LHDs were selected to represent different contexts and access to referral hospitals. A qualitative design was selected as it allows rich data regarding beliefs and experiences to be collected, with participants’ perspectives situated within the broader social contexts of rural and remote communities [35].

Participants in this study were GP-VMOs and rural and remote patients who presented to the ED of selected hospitals and received treatment or medical care from GPs working in hospital care. Involving both providers and patients as participants allowed the researchers to collect comprehensive data related to access to care, provision and perceived quality of care, and health outcomes from multiple perspectives. An invitation email was sent to 11 GP-VMOs from eight general practices in eight towns. 

Five expressions of interest were received; however, two subsequently withdrew from the study. A total of three GP-VMOs from three practices were successfully interviewed. Patients were recruited from the same eight practices. A total of 26 expressions of interest were received from four practices. Two were excluded due to lack of contact details, and the remaining 24 were followed-up through email and phone calls. Twelve patients did not respond, and two withdrew from the study. Ten patients from three practices were successfully interviewed.

Data were collected through in-depth interviews to explore experiences and perspectives of GP-VMOs and patients. Interviews were conducted remotely using video (Zoom) (n = 1) and telephone (n = 12). All interviews were conducted in English, and each took between 30–60 min to complete; conversations were audio-recorded and detailed notes were taken with participant permission. 

Using an interview guide, GP-VMO questions involved the following core topics: socio-demographic characteristics, motivations for and experiences of working in rural and remote areas, perspectives on effects of rural GP-VMOs on quality of care and health outcomes, and future aspirations. For patients, the interview guide covered the following key topics: access to GP and hospital care, experiences in accessing local ED services, perspectives and experiences of being treated by GP in a hospital, and socio-demographic characteristics. Interview schedules (for patient and GP-VMO) can be found in the Appendix A.

Data analysis: All interviews were transcribed using a third-party transcription service, subject to appropriate confidentiality provisions. The transcripts were reviewed and checked for accuracy and completeness by two authors (RK and NS). The data were analysed using thematic analysis, which identified, categorised, and reported themes within the data [36]. Data analysis was performed using NVivo-11 software. The first and second authors read the content several times while observing the general patterns in participant responses. 

Codes were developed inductively. RK developed codes from all transcripts, and these codes were validated by NS by independently reviewing a sample of the interview transcripts. Coding were compared for consistency, and any discrepancies were resolved through discussion between the two reviewers. Throughout this process, recurrent ideas were coded, and codes were sorted into categories and themes. Key themes are descriptively presented using a narrative approach supported by direct quotes from the transcripts.

Ethics approval: The study was approved by the Human Research Ethics Committees (HREC) Australia National University (Protocol Number 2020/530).

## 3. Results

### 3.1. Demographic Characteristics of Research Participants

We conducted 13 interviews with 10 patients and three GP-VMOs. All patient participants (n = 10) were >65 years of age; half (n = 5) were female and six had a tertiary level of education (TAFE or University). Most patient participants were either retired (five patients) or unemployed (two patients), and four reported living with a chronic condition. Half (n = 5) visited their GP between seven and twelve times per year, while four visited less often, and one visited more frequently. All GP-VMO participants (n = 3) were male, aged over 55 years of age, with previous ED experience and additional training either in surgery or trauma. Two were overseas trained, while only one had family responsibilities.

### 3.2. Perceived Roles of GP-VMOs in Improving the Perceived Quality of Care

This study identified three perceived roles in which GP-VMOs influenced the perceived quality of health care provided to rural and remote patients. These included improving the continuity of care, cultivating trust from local communities, and promoting integrated health services and enhancing patient satisfaction.

#### 3.2.1. Improving Continuity of Caring Relationships

Continuity of care is concerned with the consistency of care over time and across settings. From a patient perspective, emphasis is given to the on-going relationship between the health provider and the patient [37]. From a health service perspective, continuity also refers to effective management and coordination of care between different agencies or levels of care to provide responsive services [38]. Research participants described the importance of continuity of care and how GP-VMOs can play a significant role in ensuring it occurs. 

Patients in this study identified improved communication across health providers, improved interpretation, and greater understanding regarding clinical and personal circumstances of the patients, leading to enhanced trust between patients and providers. Furthermore, GP-VMOs were able to follow patients from GP surgery to the local hospital, and similarly, to monitor patients at primary care level after hospitalization. This created better linkages between primary and secondary care that promote integrated services. Some study participants felt that these circumstances also improved the quality of referral to tertiary hospitals and released beds in those hospitals for more serious conditions.

“*The fact that he knows my medical history, he knows me, he knows my temperament… He knows my medications; he follows what the specialists say. He has full knowledge of my background …*” [*Patient-05*]

“*Because most of my people, they follow me in the hospital… I know the continuity of care… They should try to keep the GP-VMOs working in the same location, the continuity of care is very good…*” [*GP-01*]

GP-VMO participants described that prior clinical knowledge about patients and social understanding about patient circumstances were essential elements of providing patient-centred and holistic care. In practice, therapeutic relationships encompassed personal understanding, mutual trust and confidence, effective communication, active listening, and sometimes extended to responding to non-clinical aspects patients’ lives. Similarly, GPs were aware of the multi-morbidities and poly-pharmacies of their patients, especially in the context of the ageing population in rural and remote areas.

“*Twice I took the patient from the hospital and brought them back... The patient had no transport and I said I can drop you to* [*large regional town*] *and bring your family and then you can follow up with the cardiologist*” [*GP-01*]

“*That is apart from varicose veins and you know things you get with age like osteopenia and that type of thing… and I have very high blood pressure… he keeps an eye on me…*” [*Patient-05*]

GP-VMOs in this study brought their contextualised knowledge of patients and the community, including understanding of social determinants of health, to clinical encounters, enabling greater continuity across levels of care and the life-course of patients.

#### 3.2.2. Cultivating Trust from Local Communities in Health Systems

Trust is critical in the delivery of health services, and is fundamental in effective treatments and for patient-centred care [34]. Trust influences the relationships between health providers and patients. The availability of GP-VMOs in rural and remote areas was described by our participants as a facilitator of comprehensive and holistic care, which is fundamental for promoting trust in health systems. Participants further highlighted that trust develops over time, resulting from ongoing interactions between GPs and their patients, especially when the GP is respectful, skillful, and living within local communities. Trust emerged when patients perceived they were treated with quality care and resulted in good health outcomes.

“*…you feel a certain warmth towards your own GP if you’ve been lucky enough to have one ongoing…Now that appeals to us greatly…, because you’re having continuity… one whole doctor for a lengthy period of time which is fantastic*” [*Patient-08*]

“*…when I first came here there was a doctor who had been here for years. The whole community felt a sense of confidence when you’re going to see Dr [Name]. He knew who you were and that’s important, just the fact that he’s familiar with you, that really makes you feel good, especially when you’re ill.*” [*Patient-07*]

Similarly, trust in health systems among rural and remote patients was often a result of the grounded, comprehensive clinical knowledge of GP-VMOs about patients. Patients were assured that their GPs were able to provide the best care possible for them because their GP understood their clinical conditions better than locums or fly-in/fly-out doctors. Participants signalled that trust in accessing hospital care was a result of having doctors who were familiar with patients both clinically and personally. 

Some degree of personal relationship between patients and GPs in hospital encounters was important in promoting trust, as well as in improving the patient experience and satisfaction. While our participants acknowledged that having competent and respectful locum doctors was very helpful and better than having no doctors at all, they preferred someone who was familiar with their clinical conditions and understood their social context. Comprehensive clinical knowledge, social understanding, and personal relationships between GP-VMOs and patients promoted trust between patients and providers, as well as in health systems.

“*We have got a good rapport with the doctor… we can communicate really well … He communicates to me and tells me what’s wrong or what he thinks I should do, and I can understand*” [*Patient-01*]

“*… I guess it’s always good if you can see your own GP but if you’re sick enough you don’t really care who you see*” [*Patient-03*]

“*It would be great because he [GP-VMO] knows your history… When you get the locum you have never dealt before, you have got to explain everything again and again…*” [*Patient-04*]

#### 3.2.3. Promoting Integrated Health Services and Patient Satisfaction in Rural and Remote Areas

Given that GP-VMOs have comprehensive clinical knowledge and social understanding concerning the patients, and familiarity with the local or regional healthcare landscape, GP-VMOs are in a good position to provide a rapid response to the health needs of rural and remote patients. Two GP-VMOs who participated in our study live within rural communities, while one resides in a nearby regional town. As well as being part of the social fabric, living locally within rural communities also means a closer proximity between their GP surgery and hospital location. 

This is important during on-call sessions, especially for emergency conditions presenting at night. Similarly, GP-VMOs may have established relationships with other services and providers and are therefore able to better navigate the system to provide responsive care for patients. From the patient perspective, the involvement of GPs in performing specific treatments in local hospitals can reduce the burden of travelling and limit disruption to patients’ lives, allowing care to be delivered closer to their home, allowing family members to contribute to care for patients, and improving mental well-being—which all contributed to healing.

“*… he’d obviously done something to his right shoulder… he was actually from* [*rural town*]*, so I basically rang [rural town] and said “Look, this guy is from* [*rural town*]*, his mother actually works in the hospital, he needs an X-ray”. It was about five o’clock. I said “Look, I’ll give him some pain relief, can you X-ray him?*” [*GP-03*]

“*… most people in these towns don’t want to have to fly out, most of them want to stay close to home, to their family and the area that they know… they’d much rather have their babies in the local area, they’d much rather be operated on in the local area if possible…*” [*GP-02*]

Our study confirmed that interpersonal skills were central to patient satisfaction. Respectful and informative communication between the patient and GP was instrumental in regulating patient emotions, facilitating their understanding of medical information, and providing clarity about patient needs and expectations. Empathy, responsiveness, and assurance were key attributes of perceived quality of care and contributed to improved patient satisfaction.

Study participants expressed high levels of satisfaction when waiting times were shorter than anticipated. Some reported that comprehensive clinical knowledge and social understanding could speed up the duration of hospital visits as well as subsequent follow-up consultations in general practice. Patients with positive outcomes, especially those who experienced acute and life threatening conditions, reported high levels of satisfaction after being cared for by their GP at the hospital, signalling that patient satisfaction and health outcomes are intricately and bi-directionally connected. Similarly, from a patient perspective, being treated by locum doctors sometimes led to perceptions of reduced quality of care. For example, it increased the duration of visits because patients had to recount their history. Additionally, patients in this study felt that the nature of locum consults was often superficial and lacked a personal ‘touch’ in comparison to GP-VMO consultations.

“*… they [GP-VMOs] know they saw you… so when you do your follow-up they have known that you’ve done that…, and so when you go for a follow-up with the GP it is really a catch up since the ED visit … so the history taking is much shorter…*” [*Patient-09*]

“*… if you get a locum come in and they have got no idea what’s wrong with you, or what you have had or your medical history… you have got to sit and tell them, a quarter of an hour and then they are saying well I have got nine more patients waiting…*” [*Patient-01*]

## 4. Discussion

This study advances current knowledge about the role of GPs who work across primary and secondary care in promoting integrated health services for rural and remote communities. A strong body of evidence has shown that low quality healthcare impedes not only improvement in health outcomes but also reduces the demand for health services [39,40]. Poor quality of care acts as a barrier to universal health coverage, independent from the access dimension [41]. 

Strategies to promote utilisation of health services and population health outcomes must adequately address quality of care issues. However, many current interventions to improve healthcare quality have focused on supply side issues and are designed to optimise service provision according to clinical guidelines [42,43]. While it is essential to address supply components, efforts to improve the quality of care should also be applied within patient-centred models [34,44] that place the perspectives of patients and communities at the core of quality improvement efforts.

In this study, rural and remote patients understood quality of care primarily through the lens of ongoing and respectful relationships with their doctors across primary and secondary care. GP-VMOs were well-positioned to improve this perceived quality, and they gained comprehensive clinical knowledge and a fuller social understanding of their patients through their ongoing interactions in primary care settings. This allowed patient-oriented care to be provided to rural patients. The ongoing care relationships between GP-VMOs and their rural patients led to four mechanisms that were considered crucial for improving the perceived quality of care: ensuring continuity of care; promoting integrated rural health care systems; cultivating trust from communities; and enhancing patient satisfaction.

Due to the limited supply of clinicians working in primary and secondary care in rural and remote areas of Australia, health needs of rural patients are often addressed through the use of locum doctors, fly-in/fly-out services, and telehealth consultations. These strategies can be fragmented and disruptive, making the continuity of caring relationships harder to establish. We found that GP-VMOs were well-positioned to improve continuity of care in rural and remote areas. They worked across primary and secondary care, enabling better linkages that promoted continuity of care and improved perceived quality of care. Furthermore, continuity of care was essential in promoting mutual trust and confidence in the health care system.

Previous studies have identified positive effects of integrated primary and secondary care in the management of chronic conditions and those with comorbid mental and physical issues, including improved clinical outcomes and the efficiency and effectiveness of care [45,46]. This is highly relevant for rural contexts with ageing populations who often have a higher prevalence of chronic conditions. Additionally, integrated care across primary and secondary health providers shows a number of system-wide advantages: promoting continuity of care, rapid response to patients’ demands, better cooperation between different levels of care, and improvement of the service delivery system [34,45,46]. 

Our study confirms that GP-VMOs are a central feature of well-integrated health services in rural and remote Australia, especially in areas with limited clinician supply, and enhance these services by working across the continuum of primary and secondary care. GP-VMOs who live locally within rural communities, understand the local and regional health service landscape and can facilitate continuing and responsive and, therefore, better health care services.

GP-VMOs, as members of smaller communities, bring an informed understanding about the population health context and risks, social and cultural perspectives and clinical knowledge of the patients into hospital care delivery. Utilising this knowledge within an integrated community-primary-secondary care model promotes the patient-centred care, which is crucial in the provision of multidisciplinary care to improve patient health outcomes [34].

Our study further highlights the importance of trust and confidence in the provision of health care services. Trust in health care systems among rural and remote patients is often a result of the grounded, comprehensive clinical knowledge about patients [34]. Personal relationships between patients and GPs are important in promoting trust in hospital encounters, as well as in improving patient experiences and satisfaction. In addition, we found that trust emerged when patients were treated with quality care resulting in positive health outcomes, especially when GP-VMOs were respectful, skilful, and living in local communities. 

Furthermore, our study indicates the importance of the inter-sectoral approach in promoting quality and timely health services for rural and remote communities, particularly across the primary and secondary health planning and deliveries. Furthermore, improving quality of health services and health outcomes for rural and remote areas requires coherent actions across different levels of governments. Improving the health outcomes of rural and remote communities requires coherent linkages between social and economic interventions with the provision of effective health services.

Given the nature of online/remote interviews due to the COVD-19 pandemic, this study faced several limitations. Patient participants who were successfully interviewed are more likely those who have better access to communication technology or perhaps higher socio-economic positions. Many of the research participants were more comfortable with phone rather than zoom interviews. Consequently, many non-verbal signs, which are valuable in face-to-face interviews could not be recorded. The current study only interviewed GPs and patients; perspectives from other health staff from hospitals and family members of the patient are not included. 

Additionally, this study included a small sample size, which limits the generalisability of our findings and may have constrained the breadth of themes and concepts elicited from the participants. In addition, GP-VMOs who participated in the study were all males; thus, no female perspective can be recorded from this study. Almost all patients who participated in the study were aged >65 years, as such, perspectives from younger patients are not adequately captured by the study. 

While our study included all patients aged over 65 years, we are not further analysed the perspectives or experiences of those who are retired and unemployed. This distinction is important given that they may have different expectations and definitions of quality health services. Further study should include this sub-analysis allowing deeper analysis of the perceived quality of care in rural and remote areas.

Our study identified three areas for future research. Further investigation detailing the social return on investment (SROI) analysis of employing GPs in hospital care for rural and remote patients may be beneficial. SROI is a framework used for understanding the social, economic, and environmental value created by a program or policy. Understanding the negative consequences of employing GPs in hospital care for rural and remote patients also requires additional research. 

Even though some studies have reported negative consequences, such as increased investigation requests and prescriptions [24] and potential psychological stress for the clinician associated with increased workload [10], these require further exploration. Lastly, studies investigating the dimensions of quality care for rural and remote patients, and whether and how this differs from urban settings is warranted to guide quality improvement strategies for rural health care.

## 5. Conclusions

Our study advances the current understanding of the role of GP-VMOs in improving the quality of services and health outcomes of rural and remote patients. We found that key roles of GP-VMOs in improving the quality of care include improving continuity of care, promoting integrated health services, cultivating trust from local communities, and enhancing the satisfaction of patients. GP-VMOs are well-positioned to improve the integration of health services for rural and remote patients, as they work across primary and secondary care creating better linkages that promote continuity of care. Employing GP-VMOs in rural hospitals enables the knowledge and sensitivity gained from their ongoing interactions with patients in primary care to be effectively utilised in the delivery of hospital care, thereby, allowing continuous, patient-centred care to be provided to rural and remote patients.

## Data Availability

The data that support this article will be shared upon reasonable request to the corresponding author.

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
