# Peer review of "Do General Practitioners in a Visiting Medical Officer Arrangement Improve the Perceived Quality of Care of Rural and Remote Patients? A Qualitative Study in Australia"

_healthcare, 2022, doi:10.3390/healthcare10061045_

Round 1
Reviewer 1 Report
Congratulations to the authors for this interesting study with implications for improving the quality of care for patients in rural and remote areas of three local health districts in New South Wales, Australia.However, I would have three minor recommendations:
1. The article should be corrected by a native English speaker to improve the expression.
2. The questions addressed to patients, but also to general practitioners (GP-VMOs) should be highlighted separately in one or two Tables to increase confidence in this study.
3. Also, even if it is a qualitative study, the way in which the questions were classified in the interviews for patients and also for GP-VMOs, but especially the classification of the answers, i.e., more precisely the way in which the statistical analysis for each category was done exactly to reach the conclusions expressed, it would be very welcome, indeed.
Author Response
Dear Reviewer,
Thank you for your highly constructive feedback to our manuscript. Please find attached in the file our comments/responses to your feedback.
Regards,
Sutarsa (On behalf of all authors)

Reviewer 2 Report
Dear Author,
It is very interesting topic, however this research is very limited regarding to the sample analysied.
Introduction - it would be valuable to provide some evidence on such research made for another countries not only literature review of Australia research.
Method- in fact you should provide questions of interview or interview guide (line 129) in detailed at least as supplementary material. It is described in too general way (lines 130-132).
Also you should be more accurate with the description of your sample - for example how many of them were retired and unemployed. You used word "most" - but is matters how many ?
"Most patient participants were either retired or unemployed, and four reported living with a chronic condition." (lines 155-156).
Moreover they should be treated separately as there is a difference between state of mind of retired and unemployed person.
Higher level of accuracy is required.
Author Response
Dear Reviewer,
Thank you for your highly constructive feedback/comments to our manuscript. Please find our comments/responses to your feedback in the attached file.
Regards,
Sutarsa (on behalf of all authors)

Reviewer 3 Report
Title
- Please consider specifying this study conducts in Australia.
Abstract
- Please include the age of participants and the date of sampling took place.
Introduction
- Few studies were used to support the research aim. The literature should be discussed the role of GP-VMOs in improving the perceived quality of care and health outcomes of patients not only in Australia but also in other countries. I would suggest that the authors refer to these articles (Aust J Rural Health. 2005;13(6):359-363; Aust J Rural Health. 2021;29(5):656-669; Hum Resour Health. 2021; 19: 126; Hum Resour Health. 2021; 19: 132; Hum Resour Health. 2019; 17: 8; Hum Resour Health. 2021; 19: 103.
- Please provide statistics on mortality and burden of diseases in rural and remote NSW. Which area suffers a greatest burden of diseases?
- More evidence on the healthcare system in rural and remote Australia is needed. Please describe the health system in terms of six core components (health workforce, health financing, health information systems, governance, health service delivery, access to medicine, vaccines and technology).
- Please clarify the determinants of health at the upstream level. Describe what makes the upstream level different from the midstream and downstream levels in rural and remote NSW.
Methods
- Line 120-124: Please define the inclusion and exclusion criteria clearly.
- Line 125-129: Data collection should be described in sufficient details.
- Line 129-135: Authors used an interview guide? How the question guide was build? Please clarify.
Results
- Quotes are long and sometimes difficult to read. Authors should consider including a table with themes/sub-themes and quotes
Discussion
- In general, this section is devoid of reference to literature (e.g., Line 380-387).
- It would be benefit to provide policy recommendations to improve the quality of services and health outcomes of patients in rural and remote NSW. Authors should look at short and long term policies with anticipated outcomes for the public health problem, with some reflection on whole of government and/or intersectoral recommendations.
Author Response
Dear Reviewer,
Thank you for your highly constructive feedback. Please find our responses to your feedback in the attached file.
Ragards,
Sutarsa (on behalf of all authors)

Round 2
Reviewer 2 Report
Dear Authors,
There is an improvement.
In fact I did not find the answer for the following comment:
"Moreover they should be treated separately as there is a difference between state of mind of retired and unemployed person"
Yes, you wrote in your answers that it was corrected and indicated the location of this corrections ("see page 4, line 155-162"). but I did not find in the text in this place.
At least you could mention about it in the discussion part as a kind of limitation. It would be good to include it.
Best regards
Author Response
Dear Reviewer,
Thank you for your constructive comments. We have addressed your feedback in the attached document.
Regards,
On behalf of all authors,
Nyoman Sutarsa

Reviewer 3 Report
Dear Authors,
Some of my comments are not sufficiently addressed.
- Line 111-112: I would expect more discussion on the factors influencing retention of remote doctors in these countries. Was any qualitative research in this area? Also, referring to only one systematic review to support the role of GP-VMOs in improving the quality of care is not enough (Line 119). These sources may add more value to the paper and I would authors suggest referring to (Aust J Rural Health. 2005;13(6):359-363; Aust J Rural Health. 2021;29(5):656-669; Hum Resour Health. 2021; 19: 126; Hum Resour Health. 2021; 19: 132; Hum Resour Health. 2019; 17: 8; Hum Resour Health. 2021; 19: 103).
- There is no mention of any national data demonstrated that rural and remote NSW experience a burden of diseases. Which area suffers a greatest burden of diseases? I feel this is an important point to add.
- Results should be re-organised so that the information is presented in a more coherent manner. Line 310-330: I would expect to see at least 3 quotes presented per theme. If there are not sufficient quotes the authors should consider whether the theme has a higher order theme that it would fit with. Please also make sure quotes are easy to read so that the reader do not feel bored when reading it (Line 233-238, Line 275-283, Line 302-309, Line 321-327, Line 360-366).
- Line 483-387: I suggest that the authors provide government and/or intersectoral policy recommendations to improve the quality of care, patient safety and health outcomes in rural and remote NSW.
Author Response
Dear Reviewer,
Thank you for your constructive feedback/comments. We have addressed your comments in the attached document.
Regards,
On behalf of all authors
Nyoman Sutarsa
